# Rim Enhancement on Contrast-Enhanced CT as a Predictor of Prognosis in Patients with Pancreatic Ductal Adenocarcinoma

**DOI:** 10.3390/diagnostics14080782

**Published:** 2024-04-09

**Authors:** Takeru Yamaguchi, Keitaro Sofue, Eisuke Ueshima, Naoki Sugiyama, Shinji Yabe, Yoshiko Ueno, Atsuhiro Masuda, Hirochika Toyama, Takayuki Kodama, Masato Komatsu, Masatoshi Hori, Takamichi Murakami

**Affiliations:** 1Department of Radiology, Kobe University Graduate School of Medicine, 7-5-2, Kusunoki-cho, Chuo-ku, Kobe 650-0017, Japan; 2Division of Gastroenterology, Department of Internal Medicine, Kobe University Graduate School of Medicine, 7-5-2, Kusunoki-cho, Chuo-ku, Kobe 650-0017, Japan; 3Division of Hepato-Biliary-Pancreatic Surgery, Department of Surgery, Kobe University Graduate School of Medicine, 7-5-2, Kusunoki-cho, Chuo-ku, Kobe 650-0017, Japan; 4Department of Pathology, Kobe University Graduate School of Medicine, 7-5-2, Kusunoki-cho, Chuo-ku, Kobe 650-0017, Japan

**Keywords:** pancreatic ductal adenocarcinoma, rim enhancement, contrast-enhanced CT

## Abstract

This study investigated the utility of imaging features, such as rim enhancement on contrast-enhanced CT (CECT), in predicting the prognosis of pancreatic ductal adenocarcinoma (PDAC). This retrospective study included 158 patients (84 men; mean age, 68 years) with pathologically confirmed PDAC. The following imaging features were evaluated on CECT by two radiologists: tumor size, tumor attenuation, and the presence of rim enhancement. Cox proportional hazards analysis was performed to identify the imaging and clinicopathological features for predicting disease-free survival (DFS) and overall survival (OS). Pathological features were compared with the presence of rim enhancement. Among the 158 patients, 106 (67%) underwent curative surgery (surgery group) and 52 (33%) received conservative treatment (non-surgery group). Rim enhancement was observed more frequently in the non-surgery group than in the surgery group (44% vs. 20%; *p* < 0.001). Rim enhancement showed significant associations with shorter DFS and OS in the surgery group (hazard ratios (HRs), 3.03 and 2.99; *p* < 0.001 and *p* = 0.003, respectively), whereas tumor size showed significant associations with shorter OS (HR per 1 mm increase, 1.08; *p* < 0.001). PDACs with rim enhancement showed significant associations with higher histological tumor grades (*p* < 0.001). PDAC with rim enhancement on CECT could predict poorer prognosis and more aggressive tumor grades.

## 1. Introduction

Pancreatic ductal adenocarcinoma (PDAC) is a lethal disease characterized by aggressive biological behavior, with a 5-year survival rate of <5% [1]. PDAC is the fourth leading cause of cancer-related deaths worldwide [2]. Surgical resection is the only potentially curative treatment for PDAC; however, other treatment options, including surgical resection, chemotherapy, radiation therapy, or a combination of these, have been attempted. Treatment strategies are selected based on disease stage and patient prognosis; thus, predicting the prognosis of patients with PDAC plays an important role in selecting an appropriate treatment strategy with reference to expected survival [3].

Multiphasic contrast-enhanced CT (CECT) is the preferred imaging modality for the detection of PDAC, evaluation of disease stage, and assessment of PDAC resectability [3]. PDAC typically presents with a gradual enhancement pattern on CECT owing to the presence of intratumoral fibrosis [4].

Preoperative imaging findings on CECT can be used to predict the prognosis of patients with PDAC. The presence of isoattenuating PDAC and higher postcontrast enhancement in the pancreatic and portal venous phase images on CECT has been associated with a better prognosis [5,6,7]. Lee et al. reported that rim-enhancing PDAC on MRI is associated with unfavorable prognosis in postsurgical patients [8]. However, the clinical utility of rim enhancement has only been clarified in a limited number of patients with pathologically confirmed R0 PDAC who are candidates for curative surgery. The significance of imaging features, such as rim enhancement on CECT, in the prediction of the prognosis of patients undergoing surgery and patients receiving conservative treatment for PDAC has not been elucidated.

Thus, this study aimed to investigate the utility of imaging features, such as rim enhancement on CECT, in predicting the prognosis of patients with PDAC.

## 2. Materials and Methods

### 2.1. Study Design and Population

This single-center, retrospective study was approved by the Institutional Review Board of our institute. The requirement for obtaining informed consent was waived owing to the retrospective nature of the study. We searched the pathological database of our institution between June 2014 and March 2020 and identified 341 patients with pathologically confirmed pancreatic cancer. The inclusion criteria were as follows: (a) patients with pathologically confirmed PDAC, (b) patients who underwent pretreatment CECT, (c) patients without a history of pancreatic surgery prior to the pretreatment CECT, (d) patients without a history of neoadjuvant chemotherapy prior to the surgery, (e) patients who had more than 6 months of follow-up after the pretreatment CECT. Patients with recurrence of PDAC after surgery (*n* = 7), patients receiving neoadjuvant chemotherapy (*n* = 39), patients with pathologies other than PDAC (*n* = 3), and patients with inadequate imaging follow-up (follow-up duration of <6 months, *n* = 134) were excluded. Thus, 158 patients, comprising 84 men and 74 women with a mean age of 68.3 ± 10.7 years, were included in the analysis (Figure 1).

### 2.2. CT Examination

Multiphasic CECT images were acquired using multidetector row CT scanners (Aquilion 64 (*n* = 99), One (*n* = 19), or Precision (*n* = 7), Canon Medical Systems, Otawara, Japan; and SOMATOM Force (*n* = 24), Siemens Healthcare, Erlangen, Germany). Table 1 presents the acquisition and reconstruction parameters of the CT scanners. An iodinated contrast material was injected into the antecubital vein using a mechanical power injector at a dose of 600 mgI/kg for a fixed duration of 30 s. Multiphasic CT images comprised unenhanced, early arterial (18–23 s), pancreatic (38–45 s), portal venous (70 s), and equilibrium (180 s) phase images. All multiphasic images were reconstructed with a slice thickness of 5 mm. The pancreatic and portal venous phase images were reconstructed with slice thickness of 0.5–0.6 mm. A bolus-tracking technique was used to acquire early arterial phase images immediately after achieving the trigger threshold.

### 2.3. Image Analysis

Two radiologists (K.S. and T.Y., with 20 and 8 years of experience in abdominal imaging, respectively) independently reviewed the CECT images acquired at initial presentation and evaluated the following imaging features: tumor size, tumor attenuation in the pancreatic and equilibrium phase images, and the presence of rim enhancement. The local resectability of each PDAC was determined according to the National Comprehensive Cancer Network (NCCN) guidelines [3]. The readers were blinded to the clinical information of the patients; however, they were not blinded to the pathologically proven diagnosis of PDAC. Rim enhancement was defined as an irregular ring-like enhancement with relative hypoattenuation in the central area. Tumor attenuation was categorized as hyperattenuation, isoattenuation, and hypoattenuation based on comparison with the surrounding normal pancreatic parenchyma in each phase image. Disagreements between the radiologists were resolved by reaching a consensus via discussion.

### 2.4. Clinical Data Collection and Survival Analysis

The following clinical data were collected from the medical records: age, sex, and serum carbohydrate antigen (CA) 19-9 levels. The following histopathological data were extracted from the pathological reports: histological grade, stromal volume, residual tumor classification, and T and N stages, according to the eighth edition of the American Joint Committee on Cancer (AJCC) staging system [9].

The medical records and follow-up images were evaluated to determine tumor recurrence and survival. For follow-up, patients underwent CECT or MRI and laboratory tests including serum CA 19-9 levels every 3–6 months after surgery. The median follow-up period was 644 days (range, 196–2815 days). Disease-free survival (DFS) was defined as the interval between the date of surgery and the date of tumor recurrence, the date of death, or the last follow-up visit. Overall survival (OS) was defined as the interval between the date of the initial CECT acquisition and the date of death or the last follow-up visit. The final evaluation was performed on 31 December 2022.

### 2.5. Histopathological Analysis

Surgical specimens with the entire tumor underwent histopathological analysis. The resected tumor specimens were fixed in 10% formalin, stained with hematoxylin and eosin, and cut into slices of 5 mm thickness. The following pathologic features were assessed: tumor size, histologic classification (well-differentiated adenocarcinoma, moderately differentiated adenocarcinoma, poorly differentiated adenocarcinoma, and adenosquamous carcinoma), stromal volume (medullary type, scant fibrous stroma; intermediate type, between medullary and scirrhous types; and scirrhous type, abundant fibrous stroma), and residual tumor classification (R0, no residual tumor; R1, microscopic residual tumor; and R2, macroscopic residual tumor) [10]. All specimens were reviewed by two pathologists (M.K. and T.K., with 10 and 5 years of experience in the pathological examination of pancreatic disease, respectively).

### 2.6. Statistical Analysis

The patients who underwent curative surgery were included in the surgery group, whereas those who received conservative treatment were included in the non-surgery group. The frequencies of the categorical variables were compared using Fisher’s exact test or χ^2^ test. Continuous variables were compared using the two-sample *t*-test if the assumption of normality was satisfied; otherwise, the Mann–Whitney U test was used. The normality of the distribution was assessed using the D’Agostino–Pearson test.

The interreader agreement for detecting rim enhancement, attenuation in the pancreatic phase images, attenuation in the equilibrium phase images, and local resectability was evaluated using weighted κ coefficients. The results were stratified qualitatively according to the scores (0.01–0.20, slight; 0.21–0.40, fair; 0.41–0.60, moderate; 0.61–0.80, substantial; and 0.81–0.99, almost perfect) [11].

Cox proportional hazard models were used to perform univariate and multivariate analyses of DFS and OS. The following variables were included in the survival analyses: age, sex, serum CA19-9 levels above the normal limit (≥37 U/mL), tumor size, tumor attenuation in the pancreatic and equilibrium phase images, and the presence of rim enhancement. Statistically significant variables in the univariate analysis were the input variables for the multivariate analysis. DFS and OS rates were estimated using the Kaplan–Meier method with log-rank analysis.

All statistical analyses were performed using MedCalc version 20 (MedCalc Software, Ostend, Belgium). A two-sided *p* value of <0.05 was considered statistically significant.

## 3. Results

### 3.1. Characteristics of the Patients and PDACs

Among the 158 PDACs included in this study, 109 (69%), 30 (19%), and 19 (12%) were considered resectable, borderline resectable, and unresectable, respectively, in the initial CECT images. Table 2 presents the patient characteristics and PDACs. Further workup (e.g., MRI, staging laparoscopy, and intraoperative ultrasound) revealed that 17 (16%) of the 109 patients with PDACs that were initially considered resectable, and 16 (53%) of the 30 patients with PDACs that were initially considered borderline resectable were unresectable. Thus, 106 patients (67%) underwent curative surgery (surgery group), and 52 (33%) received conservative treatment (non-surgery group).

### 3.2. CECT Imaging Findings

Forty-three (27%) of the one hundred and fifty-eight patients had rim-enhancing PDACs (Figure 2). Rim enhancement was observed in 25% (27/109), 33% (10/30), and 32% (6/19) of resectable, borderline resectable, and unresectable tumors, respectively. Rim enhancement was observed in 17 (52%) of the 33 patients with PDACs initially considered resectable (*n* = 17) or borderline resectable (*n* = 16) that were subsequently determined to be unresectable following further workup. Rim enhancement was observed more frequently in the non-surgery group than in the surgery group (44% (23/52) vs. 19% (20/106), *p* = 0.001). The mean size of the PDACs on the CECT image was 23.8 ± 7.1 [mm] and 37.6 ± 14.1 [mm] in the surgery and non-surgery groups, respectively. The tumor size in the non-surgery group was significantly larger than that in the surgery group (*p* < 0.001). Hypoattenuation of PDACs in the equilibrium phase images was observed more frequently in the non-surgery group than in the surgery group (*p* = 0.001). No significant difference in attenuation was observed in the pancreatic phase images (*p* = 0.10).

The interreader agreement for determining the local resectability, attenuation in the pancreatic phase images, attenuation in the equilibrium phase images, and the presence of rim enhancement was almost perfect (κ = 0.83; 95% confidence interval (95%CI), 0.74–0.90), almost perfect (κ = 0.92; 95%CI, 0.76–1.00), almost perfect (κ = 0.82; 95%CI, 0.75–0.89), and substantial (κ = 0.74; 95%CI, 0.62–0.86), respectively.

### 3.3. Histopathological Analysis of the PDACs According to Rim Enhancement

A total of 106 PDACs with surgical specimens of the entire tumor available underwent histopathological analysis (Table 3). Rim-enhancing PDACs showed significant associations with more aggressive tumor grades (*p* < 0.001) compared with non-rim-enhancing PDACs (Figure 2 and Figure 3). However, tumor size on histopathological analysis, residual tumor classification, stromal volume, and AJCC stage showed no significant associations with rim enhancement.

### 3.4. Survival Analysis

Tumor recurrence was observed in 73 (69%) of the 106 patients in the surgery group, and 44 (42%) died during the follow-up period. Univariate analysis revealed that tumor size on CECT (*p* = 0.04), hypoattenuation in the equilibrium phase images (*p* = 0.03), and rim enhancement (*p* < 0.001) showed significant associations with shorter DFS and that tumor size on CECT (*p* < 0.001) and rim enhancement (*p* < 0.001) showed significant associations with shorter OS. Age, sex, serum CA 19-9 levels above the normal limit, and hypoattenuation in the pancreatic phase images showed no significant associations with DFS or OS. Rim enhancement showed significant associations with shorter DFS and OS (hazard ratios (HRs), 3.03 and 2.99; 95%CIs, 1.66–5.54 and 1.47–6.09; *p* < 0.001 and *p* = 0.003, respectively) in the multivariate analysis. Tumor size on CECT showed significant associations with shorter OS (HR per 1 mm increase, 1.08; 95%CI, 1.03–1.13; *p* < 0.001) but no significant associations with DFS (Table 4 and Table 5). The Kaplan–Meier curves revealed significantly lower DFS and OS rates in patients with rim-enhancing PDAC than in those with non-rim-enhancing PDAC (median DFS of 186 days vs. 639 days; *p* < 0.001, and median OS of 519 days vs. 2098 days; *p* < 0.001, respectively) (Figure 4).

Thirty (58%) of the fifty-two patients in the non-surgery group died during the follow-up period. Univariate analysis revealed that serum CA 19-9 levels above the normal limit showed associations with shorter OS (*p* = 0.046). Multivariate analysis identified no significant prognostic factors, including rim enhancement on CECT images.

## 4. Discussion

The present study demonstrated that tumor size and rim enhancement on CECT were independent prognostic factors for postsurgical outcomes in patients with PDAC and that rim-enhancing PDACs were observed more frequently in patients with unresectable tumors. Rim enhancement showed significant associations with shorter DFS and OS (HRs, 3.03 and 2.99; 95%CIs, 1.66–5.54 and 1.47–6.09; *p* < 0.001 and *p* = 0.003, respectively), whereas tumor size measured on CECT showed significant associations with shorter OS (HR per 1 mm increase, 1.08; 95%CI, 1.03–1.13; *p* < 0.001). Furthermore, compared with non-rim-enhancing PDACs, histopathological analysis revealed that rim-enhancing PDACs showed associations with more aggressive tumor grades.

Tumor size has been identified as a prognostic factor for resected PDAC [12,13,14,15,16,17]. The present study also demonstrated that tumor size measured on CECT is a significant predictor of OS after curative-intent resection of PDAC; however, it is not a significant predictor of DFS. The present study included tumor size on CECT rather than on histopathological analysis in survival analysis as it aimed to predict the prognosis prior to treatment.

Prior studies have reported that preoperative CECT could predict the prognosis of PDAC [5,6]. Visually isoattenuating PDAC in the pancreatic and portal venous phase images has been associated with better survival after curative-intent surgery than hypoattenuating PDAC [5]. In the present study, however, tumor attenuation was evaluated in the pancreatic and equilibrium phase images, not in the portal venous phase images. Consequently, only three patients in the present study had tumors showing isoattenuation in the pancreatic and equilibrium phase images. Therefore, isoattenuating and hyperattenuating PDACs were combined into one category. Fukukura et al. reported that poor enhancement on pancreatic phase CECT images is associated with a poor prognosis in patients with PDAC after curative-intent surgery [6]. However, hypoattenuation in the pancreatic phase images was not a prognostic factor in the present study. This discrepancy could be attributed to Fukukura et al. defining poor enhancement according to a median CT value of 48 Hounsfield units [6]; in contrast, tumor enhancement was visually evaluated in comparison with the surrounding normal pancreatic parenchyma in the present study.

Lee et al. reported that rim-enhancing PDAC on dynamic-enhanced MRI shows associations with an unfavorable prognosis in postsurgical patients [8], which is consistent with the findings of the present study. Rim enhancement on CECT was an independent prognostic factor in patients with PDAC after curative surgery in the present study. Although MRI may possess a better ability to depict rim enhancement owing to its superior contrast resolution, CECT may be the preferred imaging modality for evaluating rim enhancement, as it is more widely used for the initial assessment of patients with PDAC. The present study demonstrated that rim enhancement was observed more frequently in patients with clinically unresectable PDACs, suggesting an association between rim enhancement and tumor aggressiveness. A previous study revealed that rim enhancement on CECT is a significant predictor of occult metastasis of PDAC, defined as metastatic disease invisible in the preoperative imaging examination and encountered intraoperatively, further demonstrating the aggressiveness of rim-enhancing PDACs [18].

Histopathological analysis revealed that rim-enhancing PDACs on CECT showed significant associations with more aggressive histological tumor grades, which is consistent with the findings of a previous study using dynamic contrast MRI [8]. This study revealed that rim-enhancing PDACs on dynamic contrast MRI exhibited significant tumor necrosis and significantly higher histological tumor grades more frequently than non-rim-enhancing PDACs. Poorly differentiated tumor grade and the presence of tumor necrosis have been reported as poor histologic prognostic factors in patients with PDAC [12,13,14,15,16,17,19,20]. Thus, the poor prognosis of rim-enhancing PDAC may be attributed to this histological association.

This study describes the utility of the imaging finding of rim enhancement on CECT as a prognostic factor for patients with PDAC. If rim-enhancing PDAC is detected on pretreatment CECT, the tumor can be more aggressive than non-rim-enhancing tumors. The utilization of non-invasive biomarkers that can help predict prognosis will aid in selecting appropriate treatment strategies and avoiding unnecessary morbidity in patients with PDAC.

The present study has some limitations. First, there may have been a potential selection bias owing to the retrospective nature of the study. Second, the study population was relatively small, particularly in the non-surgery group. Further prospective studies with larger populations must be conducted to validate these findings. Third, the presence of rim enhancement was more subjectively evaluated than the other imaging findings, which could have resulted in a worse agreement for rim enhancement. Lastly, a large number of cases were excluded owing to a lack of an adequate follow-up period and a history of receiving neoadjuvant chemotherapy, which could have introduced a selection bias. This selection process might have excluded more unresectable cases than resectable ones because those with unresectable tumors were more likely to be lost to follow-up within 6 months. Furthermore, as an increasing number of patients with PDAC are receiving neoadjuvant chemotherapy, further studies must be conducted in this population.

## 5. Conclusions

PDAC with rim enhancement on CECT is observed more frequently in patients with unresectable tumors and is an independent prognostic factor for predicting DFS and OS after curative-intent surgery. Rim enhancement of PDAC on CECT could be a predictor of tumor aggressiveness and poor prognosis, regardless of the disease stage.

## Figures and Tables

**Figure 1 diagnostics-14-00782-f001:**
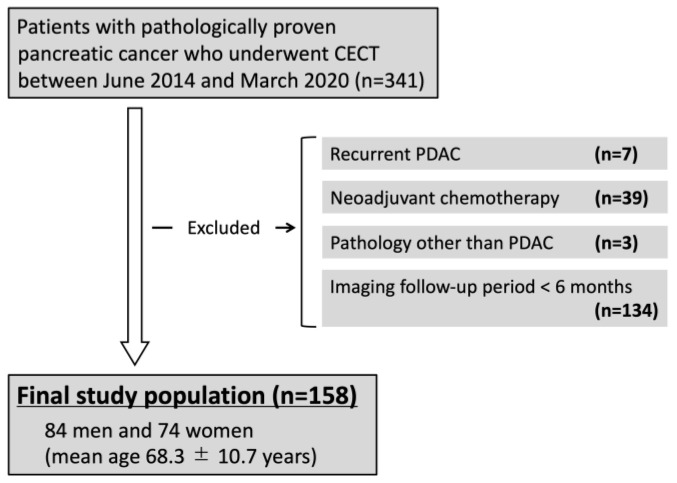
Flowchart of the study population. CECT, contrast-enhanced CT; PDAC, pancreatic ductal adenocarcinoma.

**Figure 2 diagnostics-14-00782-f002:**
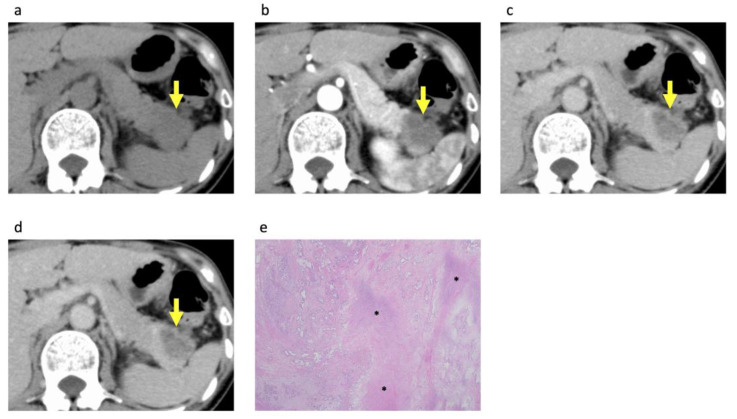
A 50-year-old woman with rim-enhancing pancreatic ductal adenocarcinoma. Transverse (**a**) unenhanced and contrast-enhanced (**b**) pancreatic phase, (**c**) portal venous phase, and (**d**) equilibrium phase CT images demonstrate a 37 mm hypovascular mass with rim enhancement in the pancreatic tail (arrows). (**e**) Photomicrograph of the pathologic specimen shows a moderately to poorly differentiated ductal adenocarcinoma with massive necrosis in the central area of the tumor (∗). (Hematoxylin–eosin stain; original magnification, ×12.5). Multiple liver metastases developed 3 months after curative-intent resection and the patient died 12 months after the surgery.

**Figure 3 diagnostics-14-00782-f003:**
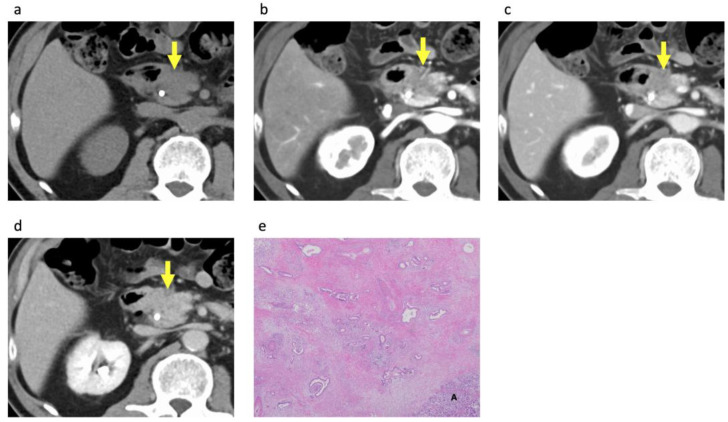
A 62-year-old man with non-rim-enhancing pancreatic ductal adenocarcinoma. Transverse (**a**) unenhanced and contrast-enhanced (**b**) pancreatic phase, (**c**) portal venous phase, and (**d**) equilibrium phase CT images demonstrate a 27 mm hypovascular mass with gradual homogeneous enhancement in the pancreatic head (arrows). Biliary stent is placed. (**e**) Photomicrograph of the pathologic specimen shows well-differentiated ductal adenocarcinoma with abundant fibrous stroma. A = remaining acini around the tumor. (Hematoxylin–eosin stain; original magnification, ×20). The patient has survived for 36 months after the surgery without tumor recurrence.

**Figure 4 diagnostics-14-00782-f004:**
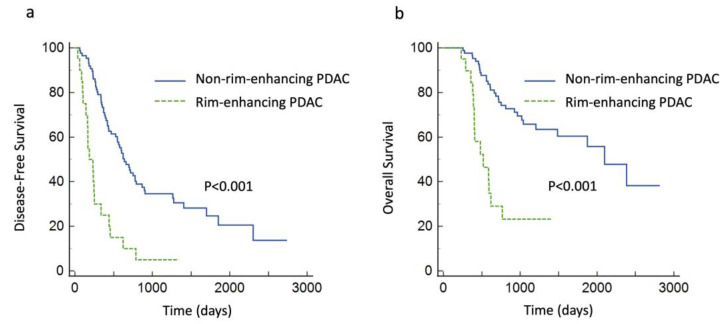
Kaplan–Meier curves show significantly lower (**a**) disease-free and (**b**) overall survival rates in patients with rim-enhancing pancreatic ductal adenocarcinoma (PDAC) than in those with non-rim-enhancing PDAC (*p* < 0.001 for both).

**Table 1 diagnostics-14-00782-t001:** CT Acquisition and Reconstruction Parameters.

	Aquilion 64	Aquilion One	Aquilion Precision	SOMATOM Force
Acquisition parameters				
Number of channels	64	320	160	192
Tube voltage (kVp)	120	120	120	120
Detector configuration (mm)	64 × 0.5	80 × 0.5	80 × 0.5	192 × 0.6
Acquisition matrix	512 × 512	512 × 512	512 × 512	512 × 512
Pitch factor	0.641	0.813	0.813	0.6
Rotation time (s)	0.5	0.5	0.5	0.5
Tube current–time product	AEC	AEC	AEC	AEC
Reconstruction parameters				
Reconstruction plane	Axial	Axial	Axial	Axial
Section thickness (mm)	5	5	5	5
Reconstruction interval (mm)	5	5	5	5
Thin-slice reconstruction parameters				
Reconstruction plane	Axial	Axial	Axial	Axial
Section thickness (mm)	0.5	0.5	0.5	0.6
Reconstruction interval (mm)	0.3	0.3	0.3	0.4

Note. Detector configuration represents the number of detector rows multiplied by the detector collimation. Thin-slice images were reconstructed from pancreatic and portal venous phase images. AEC, automatic exposure control.

**Table 2 diagnostics-14-00782-t002:** Characteristics of the Patients and PDACs.

Characteristic	Finding				
	Surgery Group		Non-Surgery Group		*p* Value
Patient					
Number	106	(67%)	52	(33%)	
Age (years)	69.1	±10.5	66.7	±9.9	0.15
Sex					0.73
Men	55	(52%)	29	(56%)	
Women	51	(48%)	23	(44%)	
CA 19-9					>0.99
>37 (U/mL)	84	(79%)	41	(85%)	
≤37 (U/mL)	22	(21%)	11	(15%)	
PDAC					
Tumor size on CECT (mm)	23.8	±7.1	37.6	±14.1	<0.001
Local resectability					<0.001
Resectable	92	(87%)	17	(33%)	
Borderline resectable	14	(13%)	16	(31%)	
Unresectable	0	(0%)	19	(36%)	
Attenuation in pancreatic phase					0.09
Hypoattenuation	99	(93%)	52	(100%)	
Iso- or hyperattenuation	7	(7%)	0	(0%)	
Attenuation in equilibrium phase					0.001
Hypoattenuation	35	(33%)	32	(62%)	
Iso- or hyperattenuation	71	(67%)	20	(38%)	
Rim enhancement					0.001
Presence	20	(19%)	23	(44%)	
Absence	86	(81%)	29	(56%)	

Note. Data are summarized as mean ± standard deviation for continuous variables or as counts (percentage) for categorical variables. CA, carbohydrate antigen; PDAC, pancreatic ductal adenocarcinoma; CECT, contrast-enhanced CT.

**Table 3 diagnostics-14-00782-t003:** Histopathologic Analysis of PDACs According to Rim Enhancement.

	Rim-Enhancing PDAC (*n* = 20)		Non-Rim-Enhancing PDAC (*n* = 86)		*p* Value
Tumor size on histopathologic analysis (mm)	34.0	±12.4	28.1	±9.5	0.06
Histologic grade					<0.001
Well-differentiated	0	(0%)	22	(26%)	
Moderately differentiated	9	(45%)	56	(65%)	
Poorly differentiated or adenosquamous carcinoma	11	(55%)	8	(9%)	
T stage					0.22
T1	1	(5%)	17	(20%)	
T2	15	(75%)	59	(69%)	
T3	4	(20%)	10	(12%)	
N stage					0.97
N0	7	(35%)	28	(33%)	
N1	9	(45%)	39	(45%)	
N2	4	(20%)	19	(22%)	
AJCC stage					0.84
I	7	(35%)	26	(30%)	
II	9	(45%)	39	(45%)	
III	4	(20%)	18	(21%)	
IV	0	(0%)	3	(3%)	
Stromal volume					0.13
Medullary type	2	(10%)	1	(1%)	
Intermediate type	14	(70%)	62	(72%)	
Scirrhous type	4	(20%)	14	(16%)	
N/A	0	(0%)	9	(10%)	
Residual tumor classification					0.48
R0	18	(90%)	72	(84%)	
R1	2	(10%)	14	(16%)	
R2	0	(0%)	0	(0%)	

Note. Data are summarized as mean ± standard deviation for continuous variables or as counts (percentage) for categorical variables. PDAC, pancreatic ductal adenocarcinoma; AJCC, American Joint Committee on Cancer; N/A, not available.

**Table 4 diagnostics-14-00782-t004:** Cox Hazard Analysis of Predictors for Disease-Free Survival in the Surgery Group.

Parameter	Univariate Analysis			Multivariate Analysis		
	Hazard Ratio		*p* Value	Hazard Ratio		*p* Value
Age (years)	0.99	(0.97–1.01)	0.42			
Sex (men)	0.97	(0.62–1.51)	0.90			
CA 19-9 (>37 U/mL)	1.05	(0.60–1.82)	0.86			
Tumor size on CECT (mm)	1.04	(1.00–1.07)	0.04	1.02	(0.98–1.06)	0.30
Hypoattenuation in pancreatic phase	1.08	(0.47–2.50)	0.86			
Hypoattenuation in equilibrium phase	1.64	(1.04–2.60)	0.03	1.19	(0.70–2.01)	0.49
Rim enhancement	3.56	(2.09–6.06)	<0.001	3.03	(1.66–5.54)	<0.001

Note. Data are presented as hazard ratios with 95% confidence intervals in parentheses. CA, carbohydrate antigen; CECT, contrast-enhanced CT.

**Table 5 diagnostics-14-00782-t005:** Cox Hazard Analysis of Predictors for Overall Survival in the Surgery Group.

Parameter	Univariate Analysis			Multivariate Analysis		
	Hazard Ratio		*p* Value	Hazard Ratio		*p* Value
Age (years)	1.00	(0.97–1.02)	0.75			
Sex (men)	1.01	(0.56–1.82)	0.98			
CA 19-9 (>37 U/mL)	1.00	(0.48–2.09)	0.99			
Tumor size on CECT (mm)	1.10	(1.06–1.15)	<0.001	1.08	(1.03–1.13)	<0.001
Hypoattenuation in pancreatic phase	2.26	(0.54–9.40)	0.26			
Hypoattenuation in equilibrium phase	1.28	(0.69–2.40)	0.43			
Rim enhancement	4.28	(2.21–8.28)	<0.001	2.99	(1.47–6.09)	0.003

Note. Data are presented as hazard ratios with 95% confidence intervals in parentheses. CA, carbohydrate antigen; CECT, contrast-enhanced CT.

## Data Availability

The data presented in this study are available on reasonable request from the corresponding author.

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
