# Peer review of "Rim Enhancement on Contrast-Enhanced CT as a Predictor of Prognosis in Patients with Pancreatic Ductal Adenocarcinoma"

_diagnostics, 2024, doi:10.3390/diagnostics14080782_

Round 1

Reviewer 1 Report

Comments and Suggestions for Authors

# Rim enhancement is hard to recognize on the presented CT. Please provide us clearer image of rim enhancement with pointing out the finding by arrows.

# Findings on CT images depends on the histopathological feature, especially interstitial tissue such as fibrosis. Authors are expected to show the relationship between the CT finding and the histopathological feature. Generally, poorly differenciated adenocarcinoma of the pancreatic cancer has medullary cell growth with slight fibrosis with degeneration. On the other hand, well differenciated adenocarcinoma of the pancreatic cancer has abundant fibrosis that shows well enhancement on the equivocal phase, because fibrosis contains much liquid, this finding is called as myxomatous change. 

Reviewer 2 Report

Comments and Suggestions for Authors

The authors aimed to add new tools to estimate the prognosis of pancreatic ductal adenocarcinoma. In their present work they present the influence of rim enhancement in the pancreatic contrast phase of CT. The study design is retrospective.

The following issues should be addressed:

in the study population a surprising 78% was considered resectable or borderline resectable. I feel there is some bias. In general about 10% of patients are regarded as resectable, mostly due to metastases. Please further explain your population.  

Rim enhancement is reported to predict poorer prognosis. Please add a statement of the pathologists if there is a histological correlate for the rim enhancement, is there a higher density of vessels? Inflammation?

How often there was a disagreement of the radiologists in categorization of the enhancement? As you state, that the kappa coefficient for rim enhancement was lower than for the other imaging characteristics, which are the difficulties and pitfalls in this technique?

In fig. 2 and fig. 3 the text doesnt fit to the images

In tab 4 and 5 instead of Hypoattetuation it should read Hypoattenuation

Line 256: as around 75% of patients experienced recurrence I would not call it "curative resection". 

Line 262. The sentence is difficult to understand. I'd prefer "in the present study" at the beginning of the sentence.

Round 2

Reviewer 2 Report

Comments and Suggestions for Authors

No further issues detected